# Effect of an educational program on the knowledge, attitudes, and practices of hospital nurses regarding evidence-based practice: A quasi-experimental study

**Omed Hamarasheed Mehammed-Ameen**[1,2]*, **Salih Ahmed Abdulla**[1,3]

1 Department of Nursing, Hawler Medical University, Kurdistan Region, Iraq, 2 Department of Community Health Nursing, College of Nursing, University of Kirkuk, Kirkuk, Iraq, 3 Nursing Department, Faculty of Nursing, Tishk International University, Erbil, Iraq

☉ These authors contributed equally to this work
* omed.mehammedameen@nur.hmu.edu.krd

## Abstract

### Background

Preparing nurses for evidence-based practice (EBP) requires developing their knowledge, skills, attitudes, and ability to apply evidence-based practice in clinical settings. This study assessed the effects of an educational program on hospital nurses' knowledge, attitudes, and practices regarding evidence-based practice.

### Methods

A quasi-experimental study with a control group and pre–post intervention assessments. The study was conducted among 116 hospital nurses in Kirkuk City, Iraq, selected using convenience sampling (58 in the intervention group and 58 in the control group). Nurses in the intervention group participated in an educational program on evidence-based practice. Data were collected at baseline and three months post-intervention using a structured self-administered questionnaire assessing knowledge, attitudes, and self-reported practice.

### Results

Nurses in the intervention group demonstrated statistically significant improvements in evidence-based practice knowledge, attitudes, and practices compared with baseline, while no significant changes were observed in the control group ($p < 0.05$). Repeated-measures ANOVA revealed a significant group × time interaction, with large effects on knowledge ($\eta^2 = 0.85$) and practice ($\eta^2 = 0.69$), and a moderate effect on attitudes toward evidence-based practice ($\eta^2 = 0.13$).

**Data availability statement:** All relevant data are within the paper and its Supporting Information files.

**Funding:** The author(s) received no specific funding for this work.

**Competing interests:** The authors have declared that no competing interests exist.

## Conclusion

The educational program was associated with significant improvements in nurses' EBP knowledge, attitudes, and self-reported practices. These findings suggest that structured EBP education can enhance nursing competencies and support the integration of evidence-based care into clinical practice.

## Introduction

Evidence-based practice (EBP) is now a fundamental part of modern nursing, integrating the best available research evidence with clinical expertise and patients' values to support informed decision-making, enhance care quality, and improve patient outcomes [1,2]. By promoting the systematic use of research findings in clinical care, EBP enables nursing practice to move beyond tradition and routine toward more accountable and effective healthcare delivery [3]. Despite its established benefits, the integration of EBP into routine nursing practice remains inconsistent across healthcare systems. Nurses frequently encounter multiple barriers, including limited knowledge and skills related to evidence appraisal, insufficient training opportunities, time constraints, and inadequate organizational support [4,5]. Furthermore, difficulties in interpreting and critically appraising research literature often lead nurses to rely on personal experience or informal peer consultation rather than on formal evidence sources, such as peer-reviewed studies and clinical guidelines [6–8]. These challenges contribute to a persistent gap between evidence generation and its application in clinical nursing practice.

To address these gaps, educational interventions and structured training programs have been widely implemented as strategies to strengthen nurses' EBP competencies across diverse healthcare settings [1,2,9]. Evidence from systematic reviews and meta-analyses consistently demonstrates that such interventions can significantly improve nurses' EBP-related knowledge, attitudes, skills, and self-efficacy.

However, the sustainability of these improvements and their translation into routine clinical practice remain ongoing challenges, particularly in healthcare systems where EBP is not firmly embedded within organizational culture or professional education [9–11]. In low- and middle-income countries, these challenges are often amplified by structural and educational constraints. In Iraq, to date, no empirical studies have evaluated the effectiveness of educational programs designed to enhance nurses' EBP competencies.

Although the Iraqi healthcare system has recently demonstrated growing institutional commitment to improving quality of care through the adoption of evidence-based standards and clinical protocols, there is still no explicit national policy mandating nurses to implement evidence-based practice (EBP) as a statutory requirement. Moreover, insufficient integration of EBP content into undergraduate nursing curricula leaves many nurses inadequately prepared to search for, critically appraise, and apply research evidence in clinical practice. As a result, EBP implementation in Iraqi healthcare settings remains limited. Therefore, there is a critical

need for context-specific evidence to determine whether structured educational interventions can effectively strengthen EBP competencies among hospital nurses in Iraq.

This study is informed by adult learning and EBP theories, which posit that nurses are more likely to adopt EBP when education enhances both their research knowledge and their attitudes toward its clinical application. Accordingly, educational programs are viewed as a central mechanism for strengthening EBP competence by fostering knowledge, positive attitudes, and evidence-informed decision-making [4,5]. The framework further assumes a close interrelationship among knowledge, attitudes, and practices, whereby improvements in understanding and attitudes support the integration of evidence into routine care. This perspective guided the design of the educational intervention, the selection of study outcomes, and the analysis of relationships among knowledge, attitudes, and practices [1,12,13].

Accordingly, this quasi-experimental study aims to evaluate the effect of a structured educational intervention on hospital nurses' knowledge, attitudes, and practices regarding EBP in Kirkuk City, Iraq. A secondary objective is to examine the relationships among these domains to determine whether improvements in knowledge and attitudes are associated with positive changes in EBP practices following the intervention. By generating context-specific evidence from an under-researched setting, this study contributes to national efforts to enhance nursing practice quality and supports the development of educational and institutional strategies for scaling effective EBP training models in resource-constrained healthcare systems.

## Materials and methods

### Design and setting of the study

A quasi-experimental design with a control group was used to examine the effect of an educational program on hospital nurses' knowledge, attitudes, and practices related to evidence-based practice in Kirkuk City, Iraq. Data were collected at baseline and three months post-intervention. Four governmental hospitals were purposively selected and assigned to either the intervention group (Azadi Teaching Hospital and Maternity and Delivery Hospital) or the control group (Kirkuk General Hospital and Paediatric Hospital). The study included bachelor-prepared nurses from various clinical units, capturing staff involved in a range of patient care activities.

### Participants and sampling

The participant flow through the study is presented in "Fig 1". The study initially targeted all bachelor-prepared nurses (N = 464) working in different clinical units across the selected hospitals. To account for potential dropouts, 120 nurses were recruited (60 intervention, 60 control) using a nonrandomized convenience sampling method. Four nurses (two per group) withdrew, resulting in a final sample of 116 (58 per group). Inclusion criteria were a bachelor's degree and willingness to participate, while nurses with less than one year of experience or those on extended leave were excluded. The sample size was calculated using G*Power software for repeated-measures ANOVA. A total of 84 nurses was required to detect a small-to-moderate effect size (f = 0.20) with 95% power at a significance level of α = 0.05. The selected effect size was informed by prior evaluations of educational interventions in EBP, which have reported comparable small-to-moderate effects on nurses' knowledge, attitudes, and practices [2,14].

### Data collection and study instruments

Several procedural steps were implemented to minimize the risk of contamination between the intervention and control groups. First, allocation was conducted at the hospital level to prevent interaction between participants from different study arms within the same institution. The participating hospitals operate under separate administrative structures, and no formal staff rotation or reassignment occurred between hospitals during the study period. Second, the educational intervention was delivered exclusively outside the hospitals at the College of Nursing, University of Kirkuk, and training materials

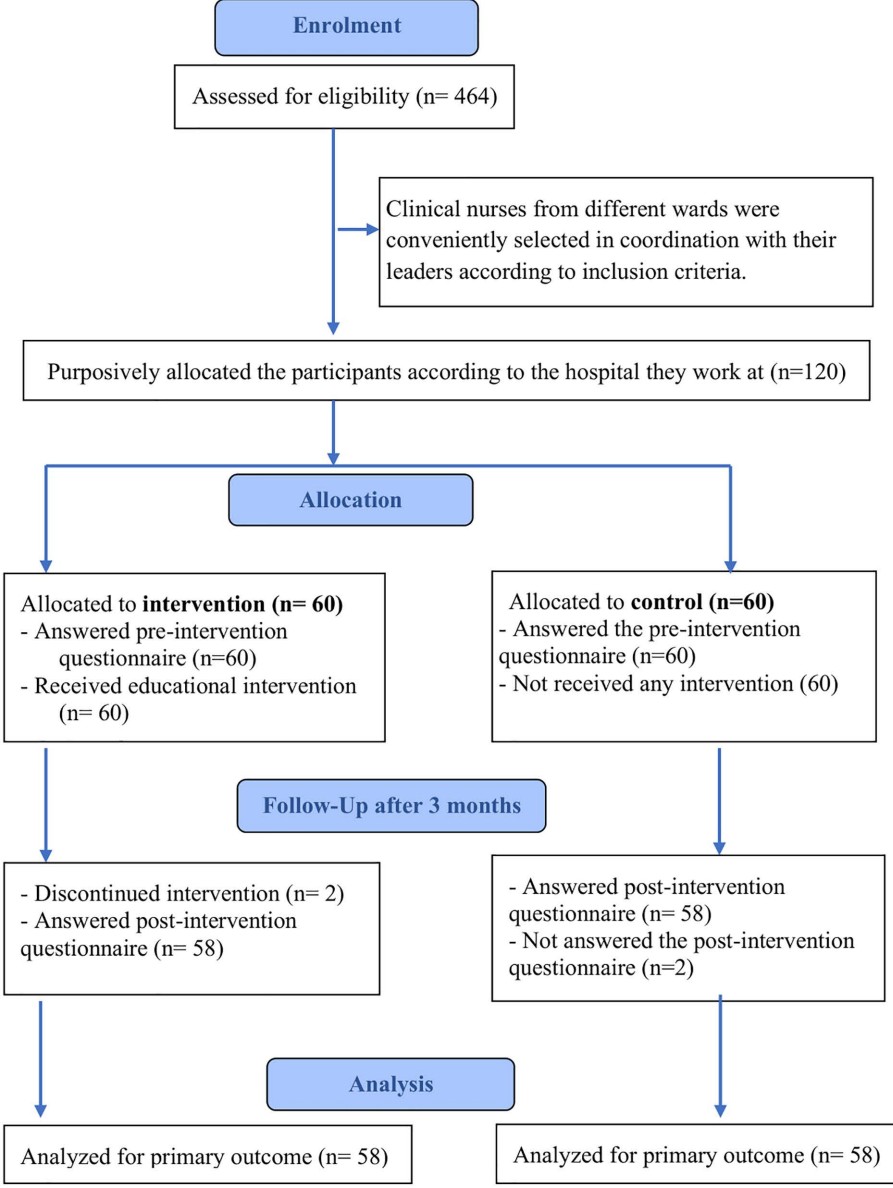

**Fig 1. Participant flow diagram of the study.** A diagram showing the enrolment, allocation, follow-up, and analysis of nurses in the intervention and control groups throughout the study period.

were not distributed beyond this site. Third, data collection in both groups was conducted within the same predefined time-frame to reduce the likelihood of delayed information transfer.

Data were collected between January and May 2025. After obtaining informed consent and explaining the study purpose, nurses in both the intervention and control groups completed a self-administered questionnaire, which required approximately 20–25 minutes to complete. Following baseline data collection, nurses in the intervention group participated in a two-week intensive educational program on evidence-based practice (EBP). Three months after completion of the intervention, participants in both groups completed the same questionnaire to assess changes in EBP-related knowledge,

attitudes, and practices. Pre- and post-intervention scores were compared to evaluate the effect of the educational program.

Data were collected using a structured self-administered questionnaire developed through an extensive literature review and adapted from two validated instruments: the Evidence-Based Practice Questionnaire for Nurses (EBPQ) by Upton and Upton [15] and the Evidence-Based Practice Competency Questionnaire–Professional Version (EBP-COQ prof©) by Ruzafa-Martinez, Lopez-Iborra [16]. Minor modifications were made to ensure contextual relevance to Iraqi hospital settings. The questionnaire comprised two sections: sociodemographic and occupational characteristics and 36 items assessing EBP competencies across three subscales including knowledge (11 items), attitudes (15 items), and practices (10 items). Knowledge items were rated on a 7-point scale ranging from 1 (poor) to 7 (good), while attitudes and practices were rated on 5-point Likert scales ranging from 1 (strongly disagree/never) to 5 (strongly agree/always).

Content validity was established by a panel of 12 experts in nursing education, research, and evidence-based practice, who evaluated the questionnaire for clarity, relevance, and comprehensiveness. Minor revisions were made based on their feedback. Internal consistency reliability was assessed through a pilot study involving 20 nurses who were not included in the main study, yielding Cronbach's alpha coefficients of 0.73 for the knowledge subscale, 0.72 for attitudes, and 0.80 for practices, indicating acceptable reliability.

## Educational program

An educational program titled "Evidence-Based Nursing Practice" was developed following an extensive review of the literature and validated prior to implementation. It was structured as a multicomponent intervention aimed at strengthening nurses' EBP competencies and was informed by established EBP models and evidence from effective educational programs [2,17].

The program was designed and delivered by one professor and two assistant professors from the College of Nursing, University of Kirkuk, all with expertise in evidence-based practice and nursing research. Participants in the intervention group were released from clinical duties during the training period and were divided into two cohorts, each completing the program over two weeks. The intervention comprised ten sessions of approximately three hours each (totalling 30 hours), integrating theoretical instruction with practical application. Participants were required to attend a minimum of nine sessions to be included in the intervention analysis. Attendance was recorded for each session to ensure compliance.

Program content followed the core EBP steps, including formulating PICO questions, searching electronic databases, critically appraising evidence, integrating findings with clinical expertise and patient preferences, and evaluating outcomes [18]. Teaching strategies combined lectures, self-directed learning, hands-on database searching, small-group workshops, case-based discussions, and mentoring, consistent with evidence supporting interactive and blended learning approaches [1,12,17]. Attendance was monitored, and formative assessments were used to reinforce learning. The control group did not receive any EBP-related education during the study period.

## Data analysis

Data were analyzed using SPSS version 27. Descriptive statistics (frequencies, percentages, means, and standard deviations) summarized the sample characteristics. Inferential analyses included chi-square and Fisher's exact tests for categorical variables and paired-sample t tests for within-group comparisons. A repeated-measures ANOVA (Time×Group) examined the effects of time, group, and their interaction on knowledge, attitudes, and practice scores. Statistical significance was set at $p \leq 0.05$, and effect sizes were reported using Partial Eta Squared ($\eta^2$), interpreted as small (0.01), medium (0.06), and large (0.14) following Cohen (1988). Pearson correlation coefficients assessed associations among the EBP domains.

## Ethical consideration

Ethical approval was obtained from the Research Ethics Committee of the College of Nursing, Hawler Medical University, Erbil, Iraq (Approval No. 246; 6 June 2024), and from the Research Committee of the Knowledge Management and Research Division, Kirkuk Governorate Health Department (Reference No. 551; 8 July 2024), permitting the study's implementation in government hospitals in Kirkuk. Authorization was also secured from hospital administrations and nursing managers, including approval for nurses in the intervention group to attend the educational program and to be released from clinical duties during the training period.

Verbal informed consent was obtained from all participants after providing a detailed explanation of the study objectives, procedures, potential risks and benefits, confidentiality, and the voluntary nature of participation, including the right to withdraw at any time without penalty. The use of verbal consent was explicitly reviewed and approved by the Research Ethics Committee due to the minimal-risk nature of the study. Consent was documented by the researcher using a standardized consent record form, which was signed and dated by the data collector to confirm that informed consent had been obtained.

Confidentiality was maintained through coded questionnaires, with no identifiable information collected, and all procedures adhered to the principles of the Declaration of Helsinki.

## Results

### Characteristics of the sample

A total of 116 participants were enrolled in the study, with 58 assigned to the control group and 58 assigned to the intervention group. Sociodemographic and occupational characteristics were collected from both groups of the study. The results show that there was no statistically significant difference in age, sex, marital status, years of experience, awareness regarding EBP, and participation in events or training related to EBP between the control group and the intervention group (Table 1).

### Changes in participants' EBP knowledge, attitudes, and practice scores after the intervention

In Table 2, paired-samples t tests were conducted separately for the control and intervention groups to determine the effects of the educational program on the participants' knowledge, attitudes, and practice scores. In terms of knowledge, the control group showed no significant difference between the pre-intervention (mean = 2.08) and post-intervention (mean = 2.10) results, with p = 0.32. The intervention group demonstrated a noteworthy increase in knowledge from pre-intervention (mean = 2.06) to the post-intervention (mean = 5.07), with a p value of less than 0.001.

In terms of attitudes, the control group showed no significant change between the pre-intervention (mean = 3.50) and post-intervention (mean = 3.56), with p = 0.18. However, the intervention group exhibited a significant positive change in attitudes, with scores increasing from the pre-intervention (mean = 3.69) to the post-intervention (mean = 4.04), with p < 0.001.

For practice, the control group showed no significant difference between the pre-intervention (mean = 2.05) and post-intervention (mean = 2.08) results, with p = 0.08. In contrast, the intervention group showed a statistically significant increase in scores from the pre-intervention (mean = 2.00) to the post-intervention (mean = 3.71), with p < 0.001.

### Effects of intervention on participants' EBP knowledge, attitudes, and practices

The repeated-measures ANOVA (Table 3) showed significant effects of the educational program on knowledge, attitudes, and practices regarding EBP. Knowledge improved significantly over time (F = 658.47, p < 0.001, $\eta^2 = 0.85$) and between groups (F = 365.96, p < 0.001, $\eta^2 = 0.76$), with a strong time–group interaction (F = 637.06, p < 0.001, $\eta^2 = 0.85$), indicating that the program's effect varied by group and time.

**Table 1. Sociodemographic and occupational characteristics of the sample (116).**

| Variables | Control (n = 58) | Intervention (n = 58) | P Value |
|---|---|---|---|
| **Age in years [Mean (SD)]** | 27.43 (± 3.42) | 27.97 (± 3.76) | 0.808 |
| **Sex [F (%)]** | | | |
| Male | 22 (37.9) | 19 (32.8) | 0.560 |
| Female | 36 (62.1) | 39 (67.2) | |
| **Marital Status [F (%)]** | | | |
| Single | 36 (62.1) | 33 (56.9) | 0.544 |
| Married | 22 (37.9) | 24 (41.4) | |
| Divorced | 0 (0) | 1 (1.7) | |
| **Hospital [F (%)]** | | | |
| Azadi teaching hospital | – | 42 (72.4) | |
| Al-Naser hospital | – | 16 (27.6) | |
| Kirkuk general hospital | 47 (81) | – | |
| Paediatric hospital | 11 (19) | – | |
| **Experience in years [Mean (SD)]** | 3.71 (± 2.12) | 4.53 (± 3.25) | 0.553 |
| **Have you ever heard about Evidence-based nursing practice? [F (%)]** | | | |
| Yes | 33 (56.9) | 29 (50) | 0.543 |
| No | 25 (43.1) | 29 (50) | |
| **Participation in events or training related to EBP? [F (%)]** | | | |
| Yes | 6 (10.3) | 4 (6.9) | 0.743 |
| No | 52 (89.7) | 54 (93.1) | |

- F = Frequency

- SD = Standard Deviation

**Table 2. Changes in participants' EBP knowledge, attitudes, and practice scores after the intervention.**

| Variables (Score range) | Pre Mean (± SD) | Post Mean (± SD) | MD (95% CI) | SMD (95% CI) | T | P value |
|---|---|---|---|---|---|---|
| **Knowledge (1–7)** | | | | | | |
| Control | 2.08 (± 0.41) | 2.10 (± 0.34) | −0.02 (−0.07, 0.02) | −0.13 (−0.39, 0.13) | −1.00 | 0.32 |
| Intervention | 2.06 (± 0.35) | 5.07 (± 0.83) | −3.00 (−3.24, −2.77) | −3.42 (−4.09, −2.74) | −26.03 | **<0.001** |
| **Attitudes (1–5)** | | | | | | |
| Control | 3.50 (± 0.31) | 3.56 (± 0.31) | −0.06 (−0.14, 0.03) | −0.18 (−0.44, 0.08) | −1.37 | 0.18 |
| Intervention | 3.69 (± 0.36) | 4.04 (± 0.31) | −0.34 (−0.46, −0.23) | −0.78 (−1.07, −0.49) | −5.96 | **<0.001** |
| **Practice (1–5)** | | | | | | |
| Control | 2.05 (± 0.60) | 2.08 (± 0.63) | −0.03 (−0.06, 0.00) | −0.23 (−0.49, 0.03) | −1.76 | 0.08 |
| Intervention | 2.00 (± 0.59) | 3.71 (± 0.51) | −1.71 (−1.92, −1.50) | −2.17 (−2.63, −1.69) | −16.48 | **<0.001** |

- SD = Standard deviation, MD = Mean Difference, SMD = Standardized Mean Difference, CI = Confidence Interval, t = paired t-test

Attitudes toward EBP also improved over time (F = 32.05, p < 0.001, $\eta^2 = 0.22$) and differed significantly across groups (F = 45.79, p < 0.001, $\eta^2 = 0.29$), with a moderate time–group interaction (F = 16.42, p < 0.001, $\eta^2 = 0.13$).

Practice scores increased markedly over time (F = 274.53, p < 0.001, $\eta^2 = 0.71$) and between groups (F = 68.97, p < 0.001, $\eta^2 = 0.38$), with a strong time–group interaction (F = 258.4, p < 0.001, $\eta^2 = 0.69$), showing that the program effects on practice were shaped by both time and group.

**Table 3. Effects of the educational program on participants' knowledge, attitudes, and practices regarding EBP (Repeated-Measures ANOVA).**

| Variable | Source | F | df | P value | n² (Effect size) |
|---|---|---|---|---|---|
| Knowledge | Time | 658.47 | 1, 114 | <0.001 | 0.85 (large) |
| | Group | 365.96 | 1, 114 | <0.001 | 0.76 (large) |
| | Time * Group | 637.06 | 1, 114 | <0.001 | 0.85 (large) |
| Attitudes | Time | 32.05 | 1, 114 | <0.001 | 0.22 (large) |
| | Group | 45.79 | 1, 114 | <0.001 | 0.29 (large) |
| | Time * Group | 16.64 | 1, 114 | <0.001 | 0.13 (medium) |
| Practice | Time | 274.53 | 1, 114 | <0.001 | 0.71 (large) |
| | Group | 68.97 | 1, 114 | <0.001 | 0.38 (large) |
| | Time * Group | 258.43 | 1, 114 | <0.001 | 0.69 (large) |

- F = F-static, df = degree of freedom, n² = partial eta squared

- Mauchly's test indicated that the assumption of sphericity was met for all within-subjects effects, W = 1.000, p > .05.

- All reported effects were statistically significant at p < 0.001

## Correlation among post intervention EBP domain scores (Intervention group)

Pearson correlation analysis showed significant positive associations among EBP domains (Table 4). Total EBP correlated very strongly with knowledge ($r = .913$, $p < .001$) and strongly with practice ($r = .853$, $p < .001$). Moderate correlations were observed with attitudes ($r = .553$, $p < .001$) and between knowledge and practice ($r = .638$, $p < .001$). Attitudes correlated moderately with practice ($r = .400$, $p = .002$), while knowledge and attitudes showed a weak but significant association ($r = .291$, $p = .027$).

## Discussion

This study is the first conducted in Iraq to evaluate the impact of a structured educational program on hospital nurses' knowledge, attitudes, and practices related to evidence-based practice (EBP). The findings demonstrate statistically significant improvements across all three domains among nurses who participated in the intervention, with no comparable

**Table 4. Pearson correlation coefficients among post intervention EBP domain scores (Intervention group).**

| Items | | Total EBP Mean score | Knowledge Mean score | Attitudes Mean score | Practice Mean score |
|---|---|---|---|---|---|
| **Intervention Group** | | | | | |
| **Total EBP Mean score** | Pearson Correlation | 1 | | | |
| | N | 58 | | | |
| **Knowledge Mean score** | Pearson Correlation | .913** | 1 | | |
| | P- value | 0.000 | | | |
| | N | 58 | 58 | | |
| **Attitudes Mean score** | Pearson Correlation | .553** | .291* | 1 | |
| | P- value | 0.000 | 0.027 | | |
| | N | 58 | 58 | 58 | |
| **Practice Mean score** | Pearson Correlation | .853** | .638** | .400** | 1 |
| | P- value | 0.000 | 0.000 | 0.002 | |
| | N | 58 | 58 | 58 | 58 |

**. Correlation is significant at the 0.01 level (2-tailed).

*. Correlation is significant at the 0.05 level (2-tailed).

changes observed in the control group. These results provide context-specific evidence supporting the effectiveness of structured EBP education. Such programs strengthen nurses' competencies and facilitate the integration of evidence-based approaches into routine clinical practice.

At baseline, nurses in both groups exhibited limited EBP knowledge and practice despite holding moderately positive attitudes toward EBP. This pattern is consistent with international evidence indicating that nurses often value EBP. However, they lack the knowledge, skills, and experiential preparation required for its application in clinical settings. Such gaps are commonly attributed to insufficient emphasis on EBP in pre-registration education and limited access to structured in-service training, particularly in resource-constrained healthcare systems [1,19,20]. These findings underscore the need for targeted educational interventions that address both theoretical understanding and practical implementation of EBP.

The observed improvements following the intervention align with evidence from experimental and quasi-experimental studies and meta-analyses demonstrating that theory-informed, multifaceted educational approaches are effective in enhancing nurses' EBP-related knowledge, attitudes, and practice behaviors. Interventions that incorporate interactive learning strategies, practical skill development, and clinically relevant applications have consistently been shown to outperform usual education or no intervention in promoting EBP competence [2,14,17,19]. The present findings extend this evidence to the Iraqi context and support the transferability of such educational models across diverse healthcare settings.

The timing of outcome assessment is important when interpreting the impact of EBP educational interventions. In this study, outcomes assessed three months after program completion showed significant improvements, consistent with evidence that benefits often emerge after a period of clinical application and reflection. Similar patterns have been reported in prior studies, where improvements were not immediate but appeared at later follow-up before declining [21]. Collectively, these findings suggest that EBP education can yield meaningful short- to mid-term gains while highlighting the importance of ongoing reinforcement and organizational support to sustain long-term improvements [14,21–23]

The significant group × time effects observed in the repeated-measures ANOVA further strengthen causal attribution to the educational program by demonstrating differential trajectories between the intervention and control groups. This pattern indicates that the observed changes were unlikely to result from maturation or external influences. Consistent with findings from randomized and controlled trials, adequately dosed, multimodal EBP training has been shown to produce sustained improvements in EBP competence and implementation over several months [11,14,21–23]. These results reinforce the methodological robustness of the intervention and its role in driving observed improvements.

The observed effect size for knowledge ($\eta^2 = 0.85$) indicates a very large proportion of variance explained by the intervention. While such magnitudes are uncommon in educational research, several factors may contribute, including low baseline knowledge, close alignment of the outcome measure with the intervention, and high sensitivity to change. Nevertheless, this large effect size should be interpreted with caution, as it may also reflect the quasi-experimental design, cluster-level influences, reliance on self-reported measures, potential response bias, and limited within-group variability, rather than the intervention effect alone.

It is important to acknowledge that the assessment of EBP practice was based on self-reported questionnaire responses, which may be subject to social desirability and response bias. Therefore, the observed improvements likely reflect perceived or reported behavior rather than objectively verified clinical practice. Future studies should incorporate objective assessment methods, such as clinical audits, direct observation, or patient-level outcome measures, to more accurately evaluate changes in evidence-based practice implementation.

Finally, the strong positive correlations observed among post-intervention knowledge, attitudes, and practice scores in the intervention group support the theoretical assumption that these domains are closely interconnected. This finding aligns with conceptual models and empirical evidence indicating that higher levels of education, structured EBP training, and positive beliefs about EBP are key determinants of nurses' evidence-based behaviors and their use of research evidence in clinical decision-making [20,24–28]. Together, these results suggest that educational interventions that simultaneously target knowledge and attitudes may be particularly effective in promoting sustained EBP practice.

## Limitations

The quasi-experimental design without randomization limits causal inference between the educational intervention and observed changes in knowledge, attitudes, and practices. Hospital-level allocation without random assignment may have introduced residual cluster-level confounding, as unmeasured institutional factors (e.g., organizational culture, leadership, staffing, and workload) may have influenced outcomes despite comparable baseline characteristics. The use of convenience sampling and inclusion of only bachelor-prepared nurses from four governmental hospitals in a single city limits generalizability, and findings may not apply to diploma nurses, private-sector hospitals, or other regions of Iraq. Volunteer bias may also have occurred, as participants may have been more motivated toward EBP, potentially inflating the intervention effects. Outcomes were based on self-reported questionnaires, which are subject to response and social desirability bias. Finally, the lack of long-term follow-up limits conclusions regarding the sustainability of the intervention's effects.

## Conclusion

This study found that participation in a structured educational program was associated with significant short-term improvements in hospital nurses' knowledge, attitudes, and self-reported practices related to evidence-based practice in Kirkuk City, Iraq. While the findings suggest that structured EBP education may enhance nurses' competencies, the results should be interpreted in light of the quasi-experimental design and reliance on self-reported practice measures. Integrating EBP-focused content into continuing professional development and undergraduate nursing curricula, supported by enabling institutional environments, may contribute to strengthening evidence-based care. Further longitudinal and randomized research incorporating objective outcome measures is warranted to confirm the magnitude and sustainability of these effects.

## Supporting information

**S1 Data. De-identified dataset underlying the findings of this study.**
(XLSX)

## Acknowledgments

The researchers appreciate all participating nurses for their time and cooperation, and they extend special thanks to the teachers of the College of Nursing at the University of Kirkuk for their contributions in designing and implementing the educational program.

## Author contributions

**Conceptualization:** Omed Hamarasheed Mehammed-Ameen, Salih Ahmed Abdulla.

**Data curation:** Omed Hamarasheed Mehammed-Ameen, Salih Ahmed Abdulla.

**Formal analysis:** Omed Hamarasheed Mehammed-Ameen.

**Methodology:** Omed Hamarasheed Mehammed-Ameen, Salih Ahmed Abdulla.

**Project administration:** Salih Ahmed Abdulla.

**Software:** Omed Hamarasheed Mehammed-Ameen.

**Validation:** Salih Ahmed Abdulla.

**Writing – original draft:** Omed Hamarasheed Mehammed-Ameen.

**Writing – review & editing:** Omed Hamarasheed Mehammed-Ameen, Salih Ahmed Abdulla.

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
