## [Decision Letter · Decision Letter 0]

16 Feb 2026

Dear Dr. Mehammed-Ameen,

Thank you for submitting your manuscript to PLOS ONE. After careful consideration, we feel that it has merit but does not fully meet PLOS ONE’s publication criteria as it currently stands. Therefore, we invite you to submit a revised version of the manuscript that addresses the points raised during the review process.

We look forward to receiving your revised manuscript.

Kind regards,

Rohit Ravi, Ph.D.

Academic Editor

PLOS One

2. In the ethics statement in the Methods, you have specified that verbal consent was obtained. Please provide additional details regarding how this consent was documented and witnessed, and state whether this was approved by the IRB.

Reviewers' comments:

Reviewer's Responses to Questions

**Comments to the Author**

1. Is the manuscript technically sound, and do the data support the conclusions?

Reviewer #1: Yes

2. Has the statistical analysis been performed appropriately and rigorously?

Reviewer #1: Yes

3. Have the authors made all data underlying the findings in their manuscript fully available?

Reviewer #1: No

4. Is the manuscript presented in an intelligible fashion and written in standard English?

Reviewer #1: Yes

Reviewer #1: This manuscript addresses an important topic and contributes context-specific evidence on evidence-based practice education among nurses in Iraq, an under-researched setting. The study design is generally appropriate, and the analyses indicate clear short-term improvements in EBP knowledge, attitudes, and self-reported practices following the intervention.

However, several issues require substantial revision before the manuscript can be considered for publication. These include clearer acknowledgment of the limitations inherent in the quasi-experimental design and non-randomized hospital allocation, greater transparency regarding the adaptation and validity of the measurement instruments, and more cautious interpretation of the unusually large effect sizes reported. In addition, the reliance on self-reported practice outcomes should be emphasized as a limitation, and conclusions should be framed accordingly.

The language is generally clear and understandable, though minor grammatical and stylistic revisions are recommended.

With these revisions, the study has the potential to make a useful contribution to the literature on EBP education in resource-constrained healthcare settings.

.

Reviewer #1: No

---

## [Author Response · Author response to Decision Letter 1]

23 Feb 2026

Response to Reviewers

Editor comments:

Response:

We thank the editor for this important reminder. We have carefully reviewed the entire submission to ensure full compliance with PLOS ONE’s style and formatting requirements.

Specifically:

• All manuscript files have been reformatted according to PLOS ONE guidelines.

2. In the ethics statement in the Methods, you have specified that verbal consent was obtained. Please provide additional details regarding how this consent was documented and witnessed, and state whether this was approved by the IRB.

Response:

We thank the editor for this important clarification. The Ethics Statement in the Methods section has been revised to provide additional details regarding the verbal consent procedure, its documentation, and IRB approval (Page 10, Line 224-232).

Reviewers’ comments:

Comments to the Author

1. Is the manuscript technically sound, and do the data support the conclusions?

Reviewer #1: Yes

2. Has the statistical analysis been performed appropriately and rigorously?

Reviewer #1: Yes

3. Have the authors made all data underlying the findings in their manuscript fully available?

Reviewer #1: No

Response: We thank the reviewer for this comment. The complete de-identified dataset underlying all findings, including the individual-level data supporting the analyses, has now been provided as Supporting Information (S1 Dataset). The Data Availability Statement has been revised accordingly to ensure full compliance with PLOS ONE policy.

4. Is the manuscript presented in an intelligible fashion and written in standard English?

Reviewer #1: Yes

5. Review Comments to the Author

Reviewer #1: This manuscript addresses an important topic and contributes context-specific evidence on evidence-based practice education among nurses in Iraq, an under-researched setting. The study design is generally appropriate, and the analyses indicate clear short-term improvements in EBP knowledge, attitudes, and self-reported practices following the intervention.

However, several issues require substantial revision before the manuscript can be considered for publication. These include clearer acknowledgment of the limitations inherent in the quasi-experimental design and non-randomized hospital allocation, greater transparency regarding the adaptation and validity of the measurement instruments, and more cautious interpretation of the unusually large effect sizes reported. In addition, the reliance on self-reported practice outcomes should be emphasized as a limitation, and conclusions should be framed accordingly.

The language is generally clear and understandable, though minor grammatical and stylistic revisions are recommended.

With these revisions, the study has the potential to make a useful contribution to the literature on EBP education in resource-constrained healthcare settings.

Response:

• We sincerely thank the reviewer for the thoughtful and constructive evaluation of our manuscript.

• We greatly appreciate the recognition of the study’s relevance and its contribution to evidence-based practice (EBP) education in an under-researched context. In response to the reviewer’s recommendations, we have undertaken substantial revisions to strengthen the manuscript.

• The Conclusion section has been revised to adopt a more cautious interpretation of the findings (Page 19-20, Line 391-400).

• The remaining recommendations are addressed in detail below in response to the major and minor comments outlined in the attached file.

6. Do you want your identity to be public for this peer review?

Reviewer #1: No

Major Comments

1. Study Design and Internal Validity

• The quasi-experimental design with hospital-level group allocation is appropriate given contextual constraints; however, non-random assignment of hospitals introduces a substantial risk of selection bias and contamination.

• Although baseline characteristics are statistically comparable, hospital-level differences (organizational culture, leadership, workload) may still explain part of the observed effects.

Recommendation: The authors should

• Explicitly acknowledge cluster-level confounding as a limitation.

• Clarify whether any steps were taken to minimize contamination (e.g., staff movement, informal knowledge sharing between hospitals).

Response:

• We agree that hospital-level allocation without randomization may introduce residual cluster-level confounding and limit internal validity. Accordingly, we have revised the Limitations section to explicitly acknowledge the potential influence of unmeasured institutional factors, which may have partially contributed to the observed effects (Page 19, Line 374-380).

• To minimize contamination, group allocation was conducted at the hospital level to prevent interaction between intervention and control participants within the same institution. The participating hospitals operate under separate administrative structures, and we assured that no formal staff rotation occurred between hospitals during the study period. Additionally, the educational intervention on the intervention group and training materials were delivered exclusively in the college of nursing /university of Kirkuk and were not accessible to control hospitals during the study period. These procedural details have now been clarified in the data collection and study instruments subsection of the methods section (Page 7, Line 147-155).

2. Sampling Strategy and Generalizability

• Convenience sampling was used, and only bachelor-prepared nurses were included.

• The sample is limited to four governmental hospitals in one city.

Recommendation: The authors should strengthen the limitations section by clearly stating that:

• Findings may not generalize to diploma nurses, private hospitals, or other regions of Iraq.

• Volunteer bias may have inflated intervention effects due to higher motivation among participants.

Response:

• We thank the reviewer for this important observation. We agree that the use of convenience sampling and restriction to bachelor-prepared nurses in four governmental hospitals may limit external validity. Accordingly, we have strengthened the Limitations section to explicitly state that the findings may not be generalizable to diploma nurses, nurses working in private-sector hospitals, or healthcare institutions in other regions of Iraq (Page 19, Line 382-386).

• We have also clarified that volunteer bias may have influenced the results, as nurses who agreed to participate may have been more motivated toward evidence-based practice, potentially inflating the observed intervention effects. These points have now been clearly incorporated into the limitations in the revised manuscript (Page 19, Line 383-386).

3. Statistical Results and Effect Sizes

• The reported effect sizes are exceptionally large (e.g., η² = 0.85 for knowledge).

• Such magnitudes are uncommon in educational interventions and raise concerns about:

Measurement sensitivity, Response bias and Ceiling effects

Recommendation: The authors should:

• Discuss why such large effects were observed.

• Acknowledge the possibility of social desirability bias, especially in self-reported practice scores.

• Consider adding confidence intervals for effect sizes or interpreting them more cautiously.

Response:

• We thank the reviewer for this important and insightful comment. We agree that the observed effect size for knowledge (η² = 0.85) is high and warrants careful interpretation.

• In the revised Discussion, we have provided a more cautious interpretation of the effect magnitude. We explain that the large effect may partially reflect low baseline knowledge levels, the structured and theory-based nature of the intervention, and the close alignment between the intervention content and the outcome measure, which may have increased sensitivity to change (Page 18, Line 349-357).

• We have also explicitly acknowledged in the Discussion section that the reliance on self-reported measures particularly for practice outcomes may have contributed to inflation of effect estimates (Page 18, Line 358-361).

4. Self- Reporting Bias

• The "Practice" domain relies on a self-administered questionnaire. Nurses may over-report

"always" performing EBP steps due to social desirability. This is a significant limitation.

Recommendation:

The discussion should clearly state that:

• Improvements in practice reflect perceived or reported behavior, not verified clinical behavior.

• Future research should incorporate audits, observations, or patient-level outcomes.

Response:

• In the discussion of the revised manuscript, we have explicitly clarified that the observed improvements in practice reflect perceived or self-reported behavior rather than objectively verified clinical performance (Page 18, Line 358-361).

• Furthermore, we have added a recommendation that future research incorporate objective evaluation methods such as clinical audits, structured observations, or patient-level outcome measures to more accurately assess the implementation of evidence-based practice (Page 18, Line 361-363).

Minor Comments

1. Abstract

• Consider adding the study design (quasi-experimental) explicitly in the first line.

• Clarify that practices are self-reported.

Response:

• We thank the reviewer for this helpful suggestion. The study design (quasi-experimental with a control group and pre–post assessments) was already specified in the Methods subsection of the Abstract. Nevertheless, we have carefully reviewed the Abstract to ensure that the design is clearly and prominently stated.

• In addition, we have revised the Abstract to explicitly clarify that the Practice domain was assessed using self-reported measures (Page 2, Line 34).

2. Introduction

• While the gap in Iraq is noted, the authors should briefly specify if there are any national nursing policies in Iraq that mandate EBP, which would further strengthen the “need” of this study.

Response:

We thank the reviewer for this constructive suggestion. In response, we have revised the Introduction to clarify the current national policy context in Iraq. Specifically, we now state that although there is increasing institutional commitment to evidence-based standards and clinical protocols, there is no explicit national policy formally mandating nurses to implement evidence-based practice as a statutory requirement. We have also highlighted the limited integration of EBP content within undergraduate nursing curricula, which may contribute to restricted implementation in practice.

This addition strengthens the contextual justification for the study and further supports the need for structured educational interventions to enhance EBP competencies among hospital nurses in Iraq (Page 4, Line 72-78).

3. Methods

• Specify whether the educational sessions were delivered by the same instructor.

• Clarify attendance thresholds (e.g., minimum sessions required for inclusion)

Response:

• We thank the reviewer for this important clarification. The Methods section has been revised to provide additional details regarding the delivery of the educational intervention and participant attendance requirements. The program was designed and delivered by one professor and two assistant professors from the College of Nursing, University of Kirkuk, all with expertise in evidence-based practice and nursing research (Page 8-9, Line 188-190).

• We have also clarified that participants were required to attend a minimum of nine out of ten sessions to be included in the intervention analysis, and attendance was recorded for each session (Page 9, Line 193-195).

4. Ethics

• Verbal consent is acceptable but briefly justify why written consent was not obtained

Response:

We thank the reviewer for this important comment. The Ethics section has been revised to clarify that verbal informed consent was used due to the minimal-risk nature of the study and because no personal identifiers were collected. The Research Ethics Committee explicitly reviewed and approved the use of verbal consent. Written consent was not deemed necessary, as participation involved completion of anonymous questionnaires and attendance at an educational program, posing no foreseeable physical or psychological risk. This clarification has been incorporated into the revised Methods section (Page 10, Line 224-230).

---

## [Decision Letter · Decision Letter 1]

18 Mar 2026

Dear Dr. Mehammed-Ameen,

Thank you for submitting your manuscript to PLOS ONE. After careful consideration, we feel that it has merit but does not fully meet PLOS ONE’s publication criteria as it currently stands. Therefore, we invite you to submit a revised version of the manuscript that addresses the points raised during the review process.

We look forward to receiving your revised manuscript.

Kind regards,

Rohit Ravi, Ph.D.

Academic Editor

PLOS One

**Journal Requirements:**

Reviewers' comments:

Reviewer's Responses to Questions

**Comments to the Author**

Reviewer #1: All comments have been addressed

2. Is the manuscript technically sound, and do the data support the conclusions?

Reviewer #1: Yes

3. Has the statistical analysis been performed appropriately and rigorously?

Reviewer #1: Yes

4. Have the authors made all data underlying the findings in their manuscript fully available?

Reviewer #1: Yes

5. Is the manuscript presented in an intelligible fashion and written in standard English?

Reviewer #1: Yes

Reviewer #1: (No Response)

.

Reviewer #1: No

---

## [Author Response · Author response to Decision Letter 2]

27 Mar 2026

Response to Reviewers

Journal Requirements:

1. We note that there is identifying data in the Supporting Information file <S1_Dataset.xlsx>. Prior to sharing human research participant data, authors should consult with an ethics committee to ensure data are shared in accordance with participant consent and all applicable local laws.

Please remove or anonymize all personal information (age), ensure that the data shared are in accordance with participant consent, and re-upload a fully anonymized data set. Please note that spreadsheet columns with personal information must be removed and not hidden as all hidden columns will appear in the published file.

Response:

Thank you for your comment. All identifiable data have been carefully reviewed and removed from the dataset. Any variables that could directly or indirectly identify participants have been deleted or replaced with non-identifiable sequential numbers. Specifically, the age column was removed, and years of experience were converted into categorical variables to minimize the risk of participant identification while preserving data utility.

The revised dataset is now fully anonymized and complies with ethical standards and data-sharing policies.

Response:

We confirm that the reviewer did not recommend citing any specific previously published works.

Response:

We have carefully reviewed the entire reference list to ensure its completeness and accuracy. All cited articles were checked for retraction status, and none were found to be retracted. Therefore, no changes to the reference list were necessary in this regard.

Reviewer comments:

Major Comments

1. Interpretation of Effect Sizes

The authors have provided a reasonable explanation for the large effect sizes observed, particularly for knowledge (η² = 0.85). However, the interpretation could be further strengthened by adopting a slightly more cautious tone.

It is recommended to explicitly acknowledge that such unusually large effect sizes may also reflect factors such as measurement alignment with the intervention, limited variability, and potential response bias, rather than purely the magnitude of the intervention effect. Adding one or two sentences of evidence that emphasize this caution would improve interpretative rigor.

Response:

We thank the reviewer for this suggestion. We have revised the Discussion to note that the large effect size for knowledge (η² = 0.85) should be interpreted with caution, as it may reflect not only the intervention but also low baseline knowledge, close alignment of the outcome measure, quasi-experimental design, cluster-level influences, reliance on self-reported measures, potential response bias, and limited within-group variability.

Minor Comments

1. Language and Style

• Minor grammatical refinement is recommended (e.g., replace “At the other hand” with “On the other hand”).

A few sentences in the Discussion section could be shortened for clarity.

Response:

• We appreciate the reviewer’s attention to language and style. The manuscript has been carefully revised to correct minor grammatical issues throughout the manuscript.

• We thank the reviewer for this suggestion. The Discussion section has been carefully revised to shorten several long sentences for improved clarity and readability, while ensuring that all key information and interpretations are fully retained (Page 17-18, Line 310-311; 314).

2. Clarity in Discussion

• Some sections of the Discussion are slightly dense; light editing for readability would enhance overall flow.

Response:

We appreciate the reviewer’s feedback. The Discussion sections and limitation has been carefully edited to improve readability and flow. Long or dense sentences have been simplified, and paragraphs have been adjusted for clarity while ensuring that all key findings and interpretations are fully preserved (Pages 17-19, Lines 328-335; 344-351; 367-379).

---

## [Editor Report · Decision Letter 2]

30 Mar 2026

Effect of an educational program on the knowledge, attitudes, and practices of hospital nurses regarding evidence-based practice: A quasi-experimental study

PONE-D-26-00727R2

Dear Dr. Mehammed-Ameen,

We’re pleased to inform you that your manuscript has been judged scientifically suitable for publication and will be formally accepted for publication once it meets all outstanding technical requirements.

Kind regards,

Rohit Ravi, Ph.D.

Academic Editor

PLOS One

Additional Editor Comments (optional):

The revision is satisfactory.
---

## [Editor Report · Acceptance letter]

PONE-D-26-00727R2

PLOS One

Dear Dr. Mehammed-Ameen,

I'm pleased to inform you that your manuscript has been deemed suitable for publication in PLOS One. Congratulations! Your manuscript is now being handed over to our production team.

Kind regards,

on behalf of

Dr. Rohit Ravi

Academic Editor

PLOS One